# Lung Cancer as a Leading Cause among Paraneoplastic Non-Bacterial Thrombotic Endocarditis: A Meta-Analysis of Individual Patients’ Data

**DOI:** 10.3390/cancers15061848

**Published:** 2023-03-20

**Authors:** Mohamed Rahouma, Sherif Khairallah, Anas Dabsha, Ismail A. M. H. Elkharbotly, Massimo Baudo, Amr Ismail, Omnia M. Korani, Mohamed Hossny, Arnaldo Dimagli, Leonard N. Girardi, Stephanie L. Mick, Mario Gaudino

**Affiliations:** 1Department of Cardiothoracic Surgery, Weill Cornell Medicine, New York, NY 10021, USA; 2Surgical Oncology Department, National Cancer Institute, Cairo University, Cairo 12613, Egypt; 3General Surgery Department, Newham University Hospital, London E13 8SL, UK; 4Department of Cardiac Surgery, Spedali Civili di Brescia, 25123 Brescia, Italy; 5Department of Medical Oncology, National Cancer Institute, Cairo University, Cairo 12613, Egypt

**Keywords:** marantic endocarditis, anticoagulation, overall survival, meta-analysis, lung cancer

## Abstract

**Simple Summary:**

Hypercoagulability has been demonstrated to have a strong association with cancer. It may result in sterile thrombotic cardiac vegetations known as “non-bacterial thrombotic” or “marantic” endocarditis. While cancer-associated NBTE is a rare entity, lung and pancreatic cancers are the most common tumor sites. Adenocarcinoma is the most common pathology, and embolization is the most common presentation. Survival is better in cases that undergo surgery and in more recent years. To prevent the potentially devastating complication of marantic endocarditis, we recommend conducting more aggressive echocardiography screening for patients with solid tumors, particularly those with adenocarcinoma of the lung and pancreas. This is based on the finding that 88.6% of patients with marantic endocarditis in this meta-analysis initially presented with embolic cerebrovascular events.

**Abstract:**

Hypercoagulability is strongly associated with cancer and may result in non-bacterial thrombotic endocarditis (NBTE). The aim of our meta-analysis was to explore the demographics and characteristics of this condition in cancer. Databases were systematically searched. The outcomes were to identify the annual trend in premortem diagnosis among the entire cohort and different subgroups and to identify differences in characteristics and survival in the considered population. A total of 121 studies with 144 patients were included. The proportion of marantic endocarditis associated with lung cancer was 0.29 (95% CI, 0.21–0.37; *p* < 0.001), that associated with pancreatic cancer was 0.19 (95% CI, 0.13–0.27; *p* < 0.001), that associated with advanced cancer stage (metastasis) was 0.69 (95% CI, 0.61–0.76; *p* < 0.001), and that associated with adenocarcinoma was 0.65 (95% CI, 0.56–0.72; *p* < 0.001). Median and 6-month overall survival (OS) were 1.3 months and 32.3%, respectively, with 6-month OS of 20.8% vs. 37.0% in lung vs. other cancers, respectively (*p* = 0.06) and 42.9% vs. 31.1% among those who underwent intervention vs. those who did not (*p* = 0.07). Cases discovered in recent years had better survival (HR = 0.98 (95% CI, 0.96–0.99; *p* = 0.003). While cancer-associated NBTE is a rare entity, lung cancers were the most common tumor site and are frequently associated with more advanced and metastatic cancer stages. The prognosis is dismal, especially among lung cancers.

## 1. Introduction

Hypercoagulability has been demonstrated to be associated with cancer. There is unlikely a single mechanism but rather a diverse set of pathophysiologic mechanisms. These include leukocytosis and thrombocytosis, the expression of tissue factor and/or phospholipid, and enhanced expression of circulating inflammatory markers. Besides patient- and treatment-related factors such as immobility, hospitalization, central venous catheters, surgery, radiation therapy, and vascular toxicity of chemotherapeutic agents may play a role [1,2]. In 1860, Armand Trousseau described the association between thrombotic events and cancer diagnosis, a phenomenon that became known as the “Trousseau sign” [3]. One of the consequences of this hypercoagulable status is the development of sterile thrombotic friable cardiac vegetations, which were first described by Zeigler R et al. in 1888 in cadavers [4] and later termed “non-bacterial thrombotic endocarditis” (NBTE) [5] or “marantic endocarditis” [6] and considered a paraneoplastic phenomenon. It is a rare entity, with an overall incidence estimated at 1.3% based on 82,676 autopsies [7]. Another study also demonstrated a low incidence between 0.9 and 1.6% in adult autopsy populations among all cases of endocarditis [8]. However, the exact prevalence of marantic endocarditis is not well established, as it is often clinically silent or an incidental finding, with most cases diagnosed postmortem [9,10,11]. Autopsy studies suggest that 75% of cases are cancer-associated, and the remainder is associated with non-cancer etiologies that are related to hypercoagulable states such as systemic lupus erythematosus [7,12]. In more recent studies, 19% of patients with solid tumors who were free from any cardiac-related symptoms were found to have sterile cardiac vegetations upon transthoracic echocardiography (TTE) screening [13]. Due to recent advances in cardiac imaging techniques, there have been marked improvements in the diagnosis and management of NBTE. One of the most important developments has been the use of transesophageal echocardiography (TEE) [9,14,15]. Furthermore, cardiac magnetic resonance imaging (MRI) was found to be very accurate in detecting clots on the heart valves, and it may be particularly useful in cases in which TEE cannot be performed or is inconclusive and to differentiate between infective and non-infective endocarditis [16]. Due to these advances, there has been an increase in the rate of the premortem diagnosis of non-bacterial thrombotic endocarditis in cancer, in addition to the monitoring of the effectiveness of treatment [17]. The first case diagnosed premortem was reported in 1976 [18]. To date, all cancer-associated NBTEs have been reported as case reports or case series; notably, there has been no systematic review of the demographics and characteristics of this condition in cancer, apart from the literature review performed by Patel et al., which included only premortem cancer and non-cancer-associated cases [19]. Addressing this knowledge gap was our aim in this systematic review and meta-analysis, which included all reported pre- and postmortem case reports or series of cancer-associated NBTE.

## 2. Materials and Methods

### 2.1. Search Strategy and Study Selection

The protocol for this review was registered with the PROSPERO registry of systematic reviews (registration number: CRD42022331513). This systematic review and meta-analysis was conducted according to the Preferred Reporting Items for Systematic Reviews and Meta-Analyses (PRISMA) guidelines [20].

In April 2022, a medical librarian performed comprehensive searches to identify studies that included all post- and premortem case reports and series of cancer-associated non-bacterial thrombotic endocarditis (NBTE) reported in the literature.

Searches were performed on the following databases: Ovid MEDLINE (ALL—1946 to present), Ovid EMBASE (1974 to present), and The Cochrane Library (Wiley). The search strategy included all appropriate controlled vocabulary and keywords for the concepts of “marantic” and “NBTE” (the full search strategy is reported in Appendix A). To limit publication bias, there were no publication date or article type restrictions on the search strategy. The reference lists of all included studies were searched to identify further articles that could potentially be recruited (i.e., backward snowballing).

Retrieved studies were screened for inclusion. Titles and abstracts were reviewed against predefined inclusion/exclusion criteria by 2 independent reviewers. Discrepancies were resolved by consensus. For final inclusion, the full text was retrieved and screened by 2 independent reviewers (A.D. and I.E.).

Articles considered for inclusion were all case reports and case series of cancer-associated non-bacterial thrombotic endocarditis “marantic” diagnosed either premortem or postmortem that included full patient data. Noncancer NBTE studies were excluded. The PRISMA flow chart is shown in Appendix A.

### 2.2. Data Extraction and Quality Assessment

Two investigators (A.D. and I.E.) performed data extraction independently. All the following variables were obtained: the presence of antiphospholipid antibodies, primary cancer organ, pathology, organ of metastasis (grouped as bone, brain, liver, lung, lymph nodes, multiple, or others), marantic presentation before the diagnosis of cancer (grouped as known cancer case or marantic as a first presentation), interval diagnosis (defined as the time between diagnosis of cancer and diagnosis of NBTE), age, chronic obstructive pulmonary disease (COPD), smoking, diabetes, obesity, dyslipidemia, hypertension, embolization event at presentation, embolic event after diagnosis, postmortem diagnosis of NBTE, position of vegetation, valve insufficiency, valve stenosis, intervention, type of surgery, presenting symptoms, and survival status and time.

In case of missing data, we reported the percentage of available data only; however, for missing survival time data, we used conditional imputation (median of alive or dead cases).

The Joanna Briggs Institute (JBI) Critical Appraisal Checklist for Case Reports was used for the quality assessment of included papers [21]. Studies were assessed using an eight-question checklist about clear description of (1) patient demographic characteristics, (2) patient history and timeline, (3) current clinical condition of the patient upon presentation, (4) diagnostic tests or assessment methods and results, (5) intervention(s) or treatment procedure(s), (6) post-intervention clinical condition, (7) adverse events (harms) or unanticipated events, and (8) takeaway lessons; each question was answered by either yes, no, unclear, or not applicable (Appendix A).

### 2.3. Outcomes of Interest

The primary outcome was to identify the annual trend in premortem diagnosis among the entire cohort.

The secondary outcomes were to identify the annual trend in premortem diagnosis among different subgroups (lung, pancreas, other gastrointestinal cancers, gynecological tumors, and others); identify differences in characteristics among (A) females vs. males, (B) lung vs. others, and (C) marantic first presentation vs. known cancer; estimate the survival difference among (A) females vs. males, (B) lung vs. others, (C) marantic first presentation vs. known cancer, (D) intervention vs. no intervention, and (E) metastatic vs. non-metastatic cases; and identify predictors of late mortality.

### 2.4. Follow-Up and Survival Analysis

Overall survival was defined as the time from diagnosis of marantic endocarditis until death from any cause. The reversed Kaplan–Meier method was used to calculate the median follow-up time.

### 2.5. Statistical Analysis

Continuous data were presented as median and interquartile range and compared using the Mann–Whitney U test or as mean and standard deviation and compared using a t-test after testing for normality. Categorical data were presented as frequency count and percentages and compared across groups using chi-square or Fisher’s test, as appropriate.

A one-sample proportions test was used to identify the probability of marantic endocarditis being associated with certain categories of certain variables, e.g., adenocarcinoma among histopathology variables.

Conditional imputation was used for cases with missed follow-up time and known follow-up status.

Cox regression was used to identify the predictors of late mortality and reported as hazard ratios (HRs) and their 95% confidence intervals (95% CIs). Variables were selected for multivariate analysis based on their statistical and clinical significance.

Overall survival was estimated using Kaplan–Meier methods and compared among groups using a log-rank test.

A staked bar plot with overlying linear regression was used to assess the annual trend of premortem diagnosis and the annual trend of cases that underwent intervention.

Data were analyzed using R version 4.1.1 (R Foundation for Statistical Computing, Vienna, Austria) within RStudio. Tableone, Survival, and Survminer packages were used.

## 3. Results

Among 829 searched studies, 121 studies [19,22,23,24,25,26,27,28,29,30,31,32,33,34,35,36,37,38,39,40,41,42,43,44,45,46,47,48,49,50,51,52,53,54,55,56,57,58,59,60,61,62,63,64,65,66,67,68,69,70,71,72,73,74,75,76,77,78,79,80,81,82,83,84,85,86,87,88,89,90,91,92,93,94,95,96,97,98,99,100,101,102,103,104,105,106,107,108,109,110,111,112,113,114,115,116,117,118,119,120,121,122,123,124,125,126,127,128,129,130,131,132,133,134,135,136,137,138,139,140,141] (published from 1962 to 2021) with 144 patients met our inclusion criteria (Table 1). The flow chart of our included studies is shown in Appendix A. Around 25% (39) of included studies were from the USA, and only 1 study was reported from China [19,22,23,24,27,29,32,33,34,39,41,43,46,47,48,57,58,59,68,69,73,76,81,84,86,88,92,93,94,100,101,105,108,111,114,115,118,131,132,136]. A list of the included studies and the quality assessment are reported in Appendix A.

According to a one-proportion test, the proportion of marantic endocarditis associated with lung cancer was 0.29 (95% CI, 0.21–0.37; *p* < 0.001), pancreatic cancer was 0.19 (95% CI, 0.13–0.27; *p* < 0.001), advanced stage (metastatic tumor) was 0.69 (95% CI, 0.61–0.76; *p* < 0.001), isolated liver metastasis was 0.23 (95% CI, 0.16–0.33; *p* ≤ 0.001), and adenocarcinoma was 0.65 (95% CI, 0.56–0.72; *p* < 0.001).

Differences in characteristics among lung vs. other cancers are reported in Table 2.

Differences in characteristics among (A) females vs. males and (B) marantic first presentation vs. known cancer are reported in Appendix A.

Around one-third (39) of the included studies was from the United States; the other third was from Japan (18), the United Kingdom (12), and Australia (8); while the last third was from the remaining 26 countries, with fewer than 4 studies each. Thus, two-thirds of the studies come from just 4 of the 30 different countries that published case reports (Appendix A).

### 3.1. Trend Analysis

There was a decrease in postmortem diagnosis with a corresponding increase in premortem diagnosis in the recent era compared to the previous era among the (A) entire group (*p* < 0.001), (B) lung cancer group (*p* = 0.006), (C) pancreatic cancer group (*p* = 0.014), (D) other gastrointestinal tract (GIT) cancers (*p* = 0.002), (E) gynecologic cancer (0.035), and (F) other cancers (*p* < 0.001) (Figure 1A–F).

Similarly, while the rate of intervention is still low (21/144, 14.6%), there was a progressive increase over the years (*p* < 0.001) (Figure 2).

### 3.2. Survival Analysis

Median follow-up time among the entire group was 5 months (95% CI: 5–8). The maximum follow-up time was 27 months. One-third of patients were alive at the end of the follow-up period.

The 1-, 3-, and 6-month overall survival among the entire group were 74.3% (95% CI: 67.4–82), 39.4% (95% CI: 31.9–48.6), and 32.3% (95% CI: 25–41.7), respectively. Median OS was 1.3 months (95% CI: 1.2–2.3). There was a trend toward poor OS among lung cancer vs. other cancers, with 1-, 3-, and 6-month overall survival of 70.1% (95% CI: 57.3–85.9), 26.0% (95% CI: 15.3–44.2), and 20.8% (95% CI: 11.2–38.4), respectively, in lung cancer vs. 76.0% (95% CI: 68.1–84.9), 44.9% (95% CI: 36–56.2), and 37.0% (95% CI: 28.0–48.9), respectively, in other cancers (log-rank *p* = 0.06). There was a trend toward better OS among those who underwent intervention vs. those who did not undergo intervention, with 1-, 3-, and 6-month overall survival of 85.7% (95% CI: 72.0–100.0), 57.1% (95% CI: 38.4–85.1), and 42.9% (95% CI: 24.1–76.1) vs. 72.6% (95% CI: 65.0–81.0), 36.7% (95% CI: 28.9–46.6), and 31.1% (95% CI: 23.6–40.9) respectively (log-rank *p* value = 0.07). There was no significant difference in OS among (A) females vs. males, (B) marantic first presentation vs. known cancer, and (C) metastatic vs. non-metastatic cases (Appendix A).

Cox multivariate analysis revealed that cases discovered in recent years were associated with better survival (HR = 0.98 (95% CI: 0.96–0.99; *p* = 0.003)), and there was a trend toward worse survival among elderly patients (HR = 1.02 (95% CI: 1.00–1.04; *p* = 0.061)) (Table 3).

## 4. Discussion

Marantic endocarditis is a type of non-bacterial endocarditis, which is an inflammation of the inner lining of the heart (endocardium) that is not due to bacterial infection. It is also called “noninfective endocarditis” or “aseptic endocarditis“. This condition is most commonly associated with cancer, particularly malignant melanoma, lung, pancreas, and breast carcinomas [17,27,57,142,143]. Marantic endocarditis can also occur in patients with other types of malignancies, as well as in patients with chronic inflammatory conditions such as rheumatoid arthritis and systemic lupus erythematosus [144].

Marantic endocarditis is characterized by the formation of small, friable, and non-bacterial vegetations on the heart valves. These vegetations are composed of fibrin, and platelets and are usually asymptomatic, but they can cause valve dysfunction and embolization if they become large [145]. In some cases, marantic endocarditis can lead to heart failure, stroke, or other serious embolic complications. The prevalence of this condition is not accurately determined. The aim of this study was to address this condition in cancer patients in order to provide a precise and conclusive recommendation regarding screening and follow-up of cancer patients for this condition.

### 4.1. Demographic, Clinical, and Pathologic Characteristics

This meta-analysis included all postmortem and premortem case reports and series of cancer-associated NBTE reported in the literature. Among 144 cases, the median age was 60 years, and 81 patients (56.6%) were females. The most common associated cancer was lung cancer (28.5%, *p* < 0.001) followed by pancreas cancer (19.4%, *p* < 0.001), gynecological malignancies (16%), and others (36.1%). This can be explained by the higher hypercoagulability state of these cancers, with the highest rates reported for venous thromboembolism and Trousseau’s syndrome [146]. Adenocarcinoma histology predominated (64.6% vs. 35.4% for other pathologies, *p* < 0.001). Marantic endocarditis was usually associated with a more advanced stage (69.3% were metastatic at initial presentation, *p* < 0.001), and the lung cancer group was the most common in subgroup analysis (87.2% (lung) vs. 62.4% (others), *p* = 0.008). Mitral and aortic valves were the most frequently affected valves (47% vs. 36.4%, respectively), with 15.9% of cases affecting more than one valve. These results are in line with the largest autopsy series to date, demonstrating more involvement of left-sided valves (mitral 43.4% vs. aortic 36% vs. multiple 13.1%). A study by Lopez et al. also demonstrated a higher prevalence of non-bacterial thrombotic endocarditis among adenocarcinoma of the lung and pancreas [7].

### 4.2. Presentation and Diagnosis

We found that there has been a decrease in the rate of postmortem diagnoses in the recent era and that NBTE is no longer considered merely a postmortem pathologic curiosity. The diagnosis of marantic endocarditis was the initial incident leading to cancer diagnosis in 66.4% of cases. In our meta-analysis, embolization events were the most common presentation (88.6%), with cerebral embolization (stroke) predominating (72.7%). This is in line with the observation by Lopez et al. that most of marantic vegetations are located on the atrial surface of the valves, with frequent embolization due to loose attachments to the endothelial surface [7]. Vegetations can also disrupt valve function [147,148], and our analysis showed that the mode of valve dysfunction was more commonly insufficiency (68.5%) than valve stenosis (10%).

Clinicians must be vigilant observers, with a high index of suspicion required to make a premortem diagnosis. Based on our results, embolic events without evidence of atherosclerotic disease, atrial fibrillation, infective endocarditis, and/or new or changed cardiac murmur should raise clinical suspicion. In three large autopsy series, 75% of marantic vegetations were less than 3 mm, and 70% were multiple [149,150,151]. Although TTE is an excellent initial modality, it has a limited ability to detect small vegetations (<3 mm) [152]; therefore transesophageal echocardiography (TEE) may be required. Additional clues for diagnosis can be found with brain MRI, which can be used to differentiate the stroke pattern of NBTE and infective endocarditis. The former usually exhibits a pattern of numerous lesions in multiple territories with variation in size, as compared to single or disseminated punctate lesions for the latter [16].

### 4.3. Management

Although the rate of intervention is still low (only 21 cases), there has been a progressive increase over years (Figure 2). As a rule, the treatment of the underlying etiology is the mainstay of treatment for non-bacterial thrombotic endocarditis. Long-term heparin therapy has been shown to be effective in preventing progression and decreasing the risk of embolic events [153,154], which stands to reason, given the hypercoagulable etiology of marantic endocarditis compared to the histological composition of platelets and fibrin. Nevertheless, other studies have demonstrated no benefit of oral anticoagulation with warfarin [155,156]. The role of surgery is unclear, with no available guidelines to guide management in this regard. Consideration of surgery should be individualized, and patients should be selected based on their performance status, degree of valve dysfunction, tumor stage, and overall prognosis. Patients with an early cancer stage with and valve dysfunction and/or recurrent embolic events, despite anticoagulation, may be considered for vegetation excision or valve replacement. In our study, among the 21 patients who underwent interventions, 16 underwent replacement, and 5 underwent excision of vegetations.

### 4.4. Prognosis

The prognosis of marantic endocarditis depends on several factors, including the underlying cause, the extent and location of the vegetations, and the patient’s overall health [15,157]. In general, the prognosis of marantic endocarditis is good, as the vegetations are usually small and do not typically cause significant damage to the heart valves. Many patients with marantic endocarditis do not experience symptoms, and the condition may be discovered incidentally on an echocardiogram or in other imaging studies [118,158]. However, as we mentioned before, in some cases, the vegetations can cause emboli that can result in serious complications such as stroke, heart attack, or organ damage [31,69,74]. These complications can lead to mortality have been reported more commonly in cancer patients [13,69,114].

Almost seventy percent of cases in this meta-analysis were associated with advanced metastatic disease, portending a poor prognosis, in line with a study by Edoute et al., who reported a higher incidence among metastatic adenocarcinoma patients [13]. Two-thirds of cases were dead at the end of the follow-up period. We could not accurately detect the exact cause of death among the included studies. However, there was a trend toward poor OS among lung cancer cases, with a poor 5-month OS of 20.8%. There was also a trend toward better OS among those who underwent interventions for NBTE treatment. This might be attributed to the removal of the embolic source, thereby preventing further embolic events and avoiding the potential risk of long-lasting anticoagulation therapy. Multivariate analysis revealed diagnosis in the recent era and younger patient age as predictors of better survival.

### 4.5. Strengths and Limitations

Our meta-analysis is the first to analyze all reported cases of cancer-associated non-bacterial thrombotic endocarditis diagnosed either pre- or postmortem and highlight the demographic and clinical characteristics, treatment outcomes, and prognosis of this rare condition using individual patients data, as all cases were reported as case reports or case series with a near-full description of the most relevant data. Unfortunately, the cause of death could not be identified in most cases. We also excluded non-English studies, which might be a source of selection bias.

## 5. Conclusions

Non-bacterial thrombotic endocarditis is a rare entity. Most of the reported cases included in this meta-analysis were cancer-associated. Lung and pancreatic cancers were the most common tumor sites. Adenocarcinoma was the most frequently associated pathology. Embolization events were the most common presentation, especially stroke. NBTE was an incidental finding, leading to cancer diagnosis in 66.4% of the reported cases, and was frequently associated with more advanced and metastatic cancer stages. Anticoagulation was the main treatment and was infrequently managed with surgery. The prognosis was dismal, especially among lung cancer cases, with a 6-month OS of 20.8%.

## Figures and Tables

**Figure 1 cancers-15-01848-f001:**
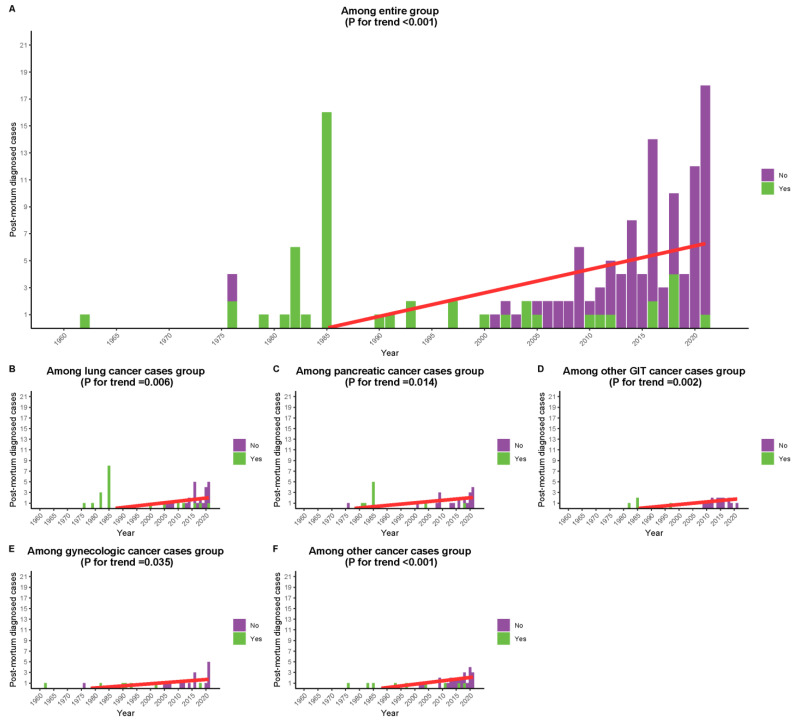
Trend of premortem diagnosis of marantic cases over the included study years among (**A**) the entire group (*p* < 0.001), (**B**) lung cancer group (*p* = 0.006), (**C**) pancreatic cancer group (*p* = 0.014), (**D**) other GIT cancers (*p* = 0.002), (**E**) gynecologic cancer (0.035), and (**F**) other cancers (*p* < 0.001).

**Figure 2 cancers-15-01848-f002:**
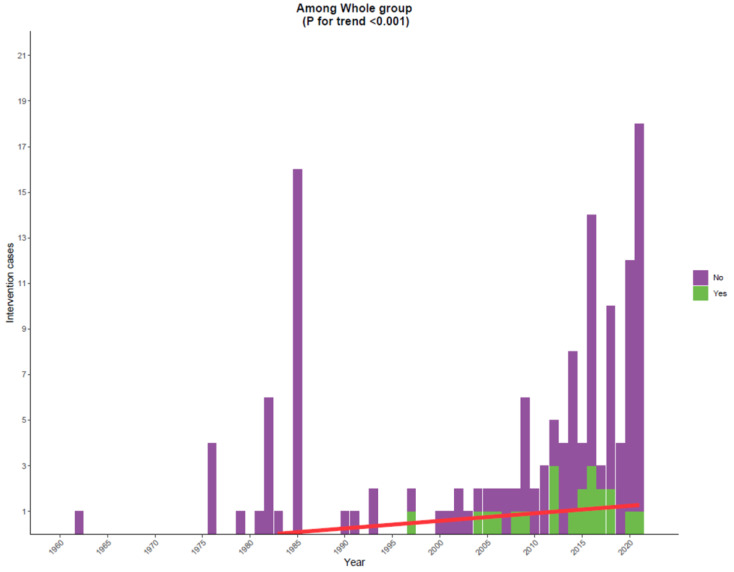
Stacked bar plot showing the number of cases that underwent intervention; while the rate of intervention is still low (21/144), there has been a progressive increase over the years.

**Table 1 cancers-15-01848-t001:** Criteria of included patients.

	Level	Overall
** *n* **		144
**Antiphospholipid antibodies (%)**	No	141 (97.9)
	Yes	3 (2.1)
**Primary cancer organ (%)**	Gynecological	23 (16.0)
	Lung	41 (28.5)
	Other GIT cancers	19 (13.2)
	Others	33 (22.9)
	Pancreas	28 (19.4)
**Lung vs. other cancers (%)**	Others	103 (71.5)
	Lung cancer	41 (28.5)
**Pathology (%)**	Adenocarcinoma	93 (64.6)
	Hematopoietic	6 (4.2)
	Others	38 (26.4)
	Sarcoma	1 (0.7)
	SCC	6 (4.2)
**Adenocarcinoma vs. other pathologies (%)**	Other pathologies	51 (35.4)
	Adenocarcinoma	93 (64.6)
**Organ of metastasis (*n* = 97) (%)**	Bone	5 (5.2)
	Brain	3 (3.1)
	Liver	23 (23.7)
	Lung	7 (7.2)
	Lymph nodes	22 (22.7)
	Multiple	25 (25.8)
	Others	12 (12.4)
**Metastasis (*n* = 141) (%)**	Non-metastatic	43 (30.7)
	Metastatic	97 (69.3)
**Marantic presentation before diagnosis of cancer (*n* = 125) (%)**	Known cancer case	42 (33.6)
	Marantic as first presentation	83 (66.4)
**Interval diagnosis (months) (mean (SD)**		6.24 (25.51)
**Age (median (IQR))**		60.00 (49.75, 66.00)
**Sex (%)**	Females	81 (56.25)
	Males	63 (43.75)
**COPD (*n* = 52) (%)**	No	49 (94.2)
	Yes	3 (5.8)
**Smoking (*n* = 56) (%)**	No	43 (76.8)
	Yes	13 (23.2)
**Diabetes (*n* = 52) (%)**	No	43 (82.7)
	Yes	9 (17.3)
**Obesity (*n* = 53) (%)**	No	52 (98.1)
	Yes	1 (1.9)
**Dyslipidemia (*n* = 54) (%)**	No	46 (85.2)
	Yes	8 (14.8)
**Hypertension (*n* = 56) (%)**	No	36 (64.3)
	Yes	20 (35.7)
**Embolization event at presentation (*n* = 132) (%)**	No	15 (11.4)
	Yes	117 (88.6)
**Embolic event after diagnosis (*n* = 125) (%)**	No	76 (60.8)
	Yes	49 (39.2)
**Incidental finding of NBTE (%)**	No	132 (91.7)
	Yes	12 (8.3)
**Postmortem diagnosis of NBTE (%)**	No	95 (66.0)
	Yes	49 (34.0)
**Position of vegetation (*n* = 132) (%)**	Aortic	48 (36.4)
	Left atrium	1 (0.8)
	Mitral	62 (47.0)
	Multiple	21 (15.9)
**Valve insufficiency (*n* = 89) (%)**	No	28 (31.5)
	Yes	61 (68.5)
**Valve stenosis (*n* = 90) (%)**	No	81 (90.0)
	Yes	9 (10.0)
**Underwent intervention (%)**	No	123 (85.4)
	Yes	21 (14.6)
**Type of surgery (%)**	Excision of vegetation	5 (3.5)
	None	123 (85.4)
	Replacement	16 (11.1)
**Presenting symptoms (*n* = 139) (%)**	Cardiological	6 (4.5)
	Neurological	96 (72.7)
	Others	7 (5.3)
	Respiratory	13 (9.8)
	Vascular	10 (7.6)
**Death (%)**	No	47 (32.6)
	Yes	97 (67.4)
**Time to death (days) (median (IQR))**		15.40 (2.73, 56.00)

**Table 2 cancers-15-01848-t002:** Different criteria among lung vs. others.

	Level	Non-Lung Cancers	Lung Cancer	*p*
**n**		103	41	
**Antiphospholipid antibodies (%)**	No	100 (97.1)	41 (100.0)	0.647
	Yes	3 (2.9)	0 (0.0)	
**Primary cancer organ (%)**	Gynecological	23 (22.3)	0 (0.0)	<0.001
	Lung	0 (0.0)	41 (100.0)	
	Other GIT cancers	19 (18.4)	0 (0.0)	
	Others	33 (32.0)	0 (0.0)	
	Pancreas	28 (27.2)	0 (0.0)	
**Pathology (%)**	Adenocarcinoma	60 (58.3)	33 (80.5)	0.004
	Hematopoietic	6 (5.8)	0 (0.0)	
	Others	34 (33.0)	4 (9.8)	
	Sarcoma	1 (1.0)	0 (0.0)	
	SCC	2 (1.9)	4 (9.8)	
**Adenocarcinoma vs. other pathologies (%)**	Other pathologies	43 (41.7)	8 (19.5)	0.02
	Adenocarcinoma	60 (58.3)	33 (80.5)	
**Organ of metastasis (n = 97) (%)**	Bone	3 (4.8)	2 (5.9)	0.258
	Brain	2 (3.2)	1 (2.9)	
	Liver	20 (31.7)	3 (8.8)	
	Lung	5 (7.9)	2 (5.9)	
	Lymph nodes	11 (17.5)	11 (32.4)	
	Multiple	15 (23.8)	10 (29.4)	
	Others	7 (11.1)	5 (14.7)	
**Metastasis (n = 141) (%)**	Non-metastatic	38 (37.6)	5 (12.8)	0.008
	Metastatic	63 (62.4)	34 (87.2)	
**Marantic presentation before diagnosis of cancer (n = 125) (%)**	Known cancer case	34 (36.2)	8 (25.8)	0.401
	Marantic as a first presentation	60 (63.8)	23 (74.2)	
**Interval diagnosis (months) (mean (SD))**		7.95 (29.56)	1.54 (3.81)	0.257
**Age (median (IQR))**		59.00 (49.00, 65.00)	61.00 (55.00, 67.00)	0.389
**Sex (%)**	F	63 (61.2)	18 (45.0)	0.118
	M	40 (38.8)	22 (55.0)	
**COPD (n = 52) (%)**	No	39 (97.5)	10 (83.3)	0.254
	Yes	1 (2.5)	2 (16.7)	
**Smoke (n = 56) (%)**	No	37 (92.5)	6 (37.5)	<0.001
	Yes	3 (7.5)	10 (62.5)	
**Diabetes (n = 52) (%)**	No	31 (77.5)	12 (100.0)	0.17
	Yes	9 (22.5)	0 (0.0)	
**Obesity (n = 53) (%)**	No	40 (97.6)	12 (100.0)	1
	Yes	1 (2.4)	0 (0.0)	
**Dyslipidemia (n = 54) (%)**	No	35 (85.4)	11 (84.6)	1
	Yes	6 (14.6)	2 (15.4)	
**Hypertension (n = 56) (%)**	No	27 (64.3)	9 (64.3)	1
	Yes	15 (35.7)	5 (35.7)	
**Embolization event at presentation (n = 132) (%)**	No	12 (12.4)	3 (8.6)	0.767
	Yes	85 (87.6)	32 (91.4)	
**Embolic event after diagnosis (n = 125) (%)**	No	55 (59.1)	21 (65.6)	0.661
	Yes	38 (40.9)	11 (34.4)	
**Incidental finding of NBTE (%)**	No	95 (92.2)	37 (90.2)	0.956
	Yes	8 (7.8)	4 (9.8)	
**Postmortem diagnosis of NBTE (%)**	No	74 (71.8)	21 (51.2)	0.031
	Yes	29 (28.2)	20 (48.8)	
**Position of vegetation (n = 132) (%)**	Aortic	29 (31.2)	19 (48.7)	0.008
	Left atrium	1 (1.1)	0 (0.0)	
	Mitral	42 (45.2)	20 (51.3)	
	Multiple	21 (22.6)	0 (0.0)	
**Valve insufficiency (n = 89) (%)**	No	23 (35.4)	5 (20.8)	0.292
	yes	42 (64.6)	19 (79.2)	
**Valve stenosis (n = 90) (%)**	No	57 (86.4)	24 (100.0)	0.131
	Yes	9 (13.6)	0 (0.0)	
**Underwent intervention (%)**	No	84 (81.6)	39 (95.1)	0.069
	Yes	19 (18.4)	2 (4.9)	
**Type of surgery (%)**	Excision of vegetation	4 (3.9)	1 (2.4)	0.096
	None	84 (81.6)	39 (95.1)	
	Replacement	15 (14.6)	1 (2.4)	
**Presenting symptoms (n = 139) (%)**	Cardiological	4 (4.2)	2 (5.6)	0.467
	Neurological	68 (70.8)	28 (77.8)	
	Others	4 (4.2)	3 (8.3)	
	Respiratory	11 (11.5)	2 (5.6)	
	Vascular	9 (9.4)	1 (2.8)	
**Death (144) (%)**	No	37 (36.3)	8 (20.0)	0.094
	Yes	65 (63.7)	32 (80.0)	
**Time to death (days) (median (IQR))**		19.60 (1.05, 56.00)	10.99 (3.50, 35.28)	0.64

**Table 3 cancers-15-01848-t003:** Predictors of late mortality using Cox regression analysis.

Variables	Hazard Ratio, 95% CI, *p* Value
**Age**	**1.021 [0.999; 1.043], 0.06157**
**Year**	**0.975 [0.959; 0.991], 0.00289**
**Underwent intervention**	1.235 [0.472; 3.229], 0.66705
**Pathology (adenocarcinoma vs. others)**	0.994 [0.560; 1.764], 0.98430
**Cancer organ (ref: lung)**	
**Pancreas**	0.928 [0.477; 1.806], 0.82582
**Other GIT cancer**	0.555 [0.210; 1.463], 0.23369
**Gynecological**	0.578 [0.180; 1.855], 0.35703
**Others**	0.724 [0.338; 1.548], 0.40459
**Organ affected by metastasis (liver vs. others)**	1.235 [0.644; 2.368], 0.52555

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
