# Peer review of "Lung Cancer as a Leading Cause among Paraneoplastic Non-Bacterial Thrombotic Endocarditis: A Meta-Analysis of Individual Patients’ Data"

_cancers, 2023, doi:10.3390/cancers15061848_

Round 1
Reviewer 1 Report
Comments: Few critical points should clarify before being accepted.
1- The manuscript lacks information about hypercoagulable status if it was formed by chemotherapies, radiotherapies or which the reason to be associated with cancer patients
2- Authors did not describe how many papers were included? the period of the publications,
3- Many English grammatical and typos errors were captured over all the text. English should have been revised thoroughly
4- In line 95, it was written (grouped as bone, brain, liver, lung, others, lymph nodes, and multiple)". others should be moved into the end).
5- The quality of figures 1 and 2 should be improved.
6- References should to be arranged according to the format style of the cancer MDPI journal.
7- Authors did not provide an explanation for why lung cancer is the most common tumor, can produce nonbacterial thrombotic endocarditis
8- Conclusion should to be improved
Author Response
Dear Reviewer,
Please find attached the revised manuscript “Lung cancer as leading cause among paraneoplastic non-bacterial thrombotic endocarditis: A meta-analysis of in-dividual patients’ data”.
All the Reviewer’s comments have been addressed. Below is a point-by-point reply to each single comment.
Reviewer 1
Comment 1: The manuscript lacks information about hypercoagulable status if it was formed by chemotherapies, radiotherapies or which the reason to be associated with cancer patients.
Answer 1: We thank the Reviewer for pointing that out. The introduction was implemented with the following: “There is unlikely a single mechanism, but rather a diverse set of pathophysiologic mechanisms. These include leukocytosis and thrombocytosis, the expression of tissue factor and/or phospholipid, and enhanced expression of circulating inflammatory markers. Besides, patient- and treatment-related factors such as immobility, hospitalization, central venous catheters, surgery, radiation therapy, and vascular toxicity of chemotherapeutic agents may play a role.”
Comment 2: Authors did not describe how many papers were included? the period of the publications.
Answer 2: Thank you for your comments. As already stated in the Result section, “Among 829 searched studies, 121 studies with 144 patients met our inclusion criteria.”. The sentence was modified to include publication year information as follows: “Among 829 searched studies, 121 studies (published from 1962 to 2021) with 144 patients met our inclusion criteria.”
Comment 3: Many English grammatical and typos errors were captured over all the text. English should have been revised thoroughly.
Answer 3: We apologize for the inconvenience. The manuscript underwent a full English revision.
Comment 4: In line 95, it was written (grouped as bone, brain, liver, lung, others, lymph nodes, and multiple)". others should be moved into the end).
Answer 4: We thank the Reviewer for pointing that out. The sentence was corrected accordingly.
Comment 5: The quality of figures 1 and 2 should be improved.
Answer 5: We apologize for the inconvenience. The quality of the figures was improved.
Comment 6: References should to be arranged according to the format style of the cancer MDPI journal
Answer 6: We apologize for the inconvenience. References were correctly formatted according to the Journal style.
Comment 7: Authors did not provide an explanation for why lung cancer is the most common tumor, can produce nonbacterial thrombotic endocarditis.
Answer 7: Thank you for your conscientious comment. The following sentence was added to the discussion: “This can be explained by the higher hypercoagulability state of these cancer, reporting the highest rates of venous thromboembolism and Trousseau’s syndrome”.
Comment 8: Conclusion should to be improved.
Answer 8: Thank you for your suggestions. The Conclusion paragraph was improved.

Reviewer 2 Report
The authors present the results of a systematic review and meta-analysis of paraneoplastic non-bacterial thrombotic endocarditis. Literature about this topic is based on many case reports, each with 1 or a few cases, and the authors try to produce a systematic synthesis of the evidence.
My comments:
The authors use the term "individual patients'data". Usually this is referred to the availability of data of each single patient. In this case, data are extracted by the publications, so I am not sure that the term "IPD" is correct. Probably this work should be labeled as "literature-based meta-analysis".
I don’t understand what the p value describing the association of marantic endocarditis with different types of tumors is referred to: “Proportion of marantic endocarditis associated with lung cancer was 0.29 [95%CI 0.21-0.37], P<0.001”. Which test was performed? A test should compare the distribution of different tumors in patients with marantic endocarditis with the distribution of different tumors in the whole population, but I suppose the authors do not have the latter information… Please clarify.
In the paragraph "recommendations" the authors state: "Given that 88.6% of these patients’ initial symptom of marantic endocarditis was embolic cerebro-vascular events, in concert with the findings of Edoute et al, that 19% of solid tumor patients demonstrate marantic endocarditis, we recommend more aggressive echo screening of patients with solid tumors, specially adenocarcinoma of lung and pancreas, to prevent this potentially devastating complication." I have two comments: first, 19% of patients with endocarditis seems quite exaggerated to me. If true, it means that in the vast majority this would remain completely asymptomatic and clinically silent. Second, I think that an aggressive screening should be recommended only if proven effective. Otherwise, recommending screening would only increase the number of echo exams, with potentially useless and time- and resource-consuming exams. I think that this paragraph should be at least reformulated or - maybe better - deleted.
Table 3: the rows "Lung vs Other cancers" are completely useless (the information is already described in the columns)
Author Response
Dear Reviewer,
Please find attached the revised manuscript “Lung cancer as leading cause among paraneoplastic non-bacterial thrombotic endocarditis: A meta-analysis of in-dividual patients’ data”.
All the Reviewer’s comments have been addressed. Below is a point-by-point reply to each single comment.
Reviewer 2
Comment 1: The authors present the results of a systematic review and meta-analysis of paraneoplastic non-bacterial thrombotic endocarditis. Literature about this topic is based on many case reports, each with 1 or a few cases, and the authors try to produce a systematic synthesis of the evidence.
Answer 1: We thank the reviewer for his/her words and time to review our manuscript.
Comment 2: The authors use the term "individual patients'data". Usually this is referred to the availability of data of each single patient. In this case, data are extracted by the publications, so I am not sure that the term "IPD" is correct. Probably this work should be labeled as "literature-based meta-analysis".
Answer 2: Thanks for your comment. As data of each included case was previously reported as a case report, so it is actually individual patients’ data.
Comment 3: I don’t understand what the p value describing the association of marantic endocarditis with different types of tumors is referred to: “Proportion of marantic endocarditis associated with lung cancer was 0.29 [95%CI 0.21-0.37], P<0.001”. Which test was performed? A test should compare the distribution of different tumors in patients with marantic endocarditis with the distribution of different tumors in the whole population, but I suppose the authors do not have the latter information… Please clarify.
Answer 3: Thanks for your thoughtful comment. As we specified in the statistical section: We used “one-sample proportions test” to identify the probability of marantic endocarditis to be associated with certain categories in certain variables e.g., adenocarcinoma among histopathology variables.
Comment 4: In the paragraph "recommendations" the authors state: "Given that 88.6% of these patients’ initial symptom of marantic endocarditis was embolic cerebrovascular events, in concert with the findings of Edoute et al, that 19% of solid tumor patients demonstrate marantic endocarditis, we recommend more aggressive echo screening of patients with solid tumors, specially adenocarcinoma of lung and pancreas, to prevent this potentially devastating complication." I have two comments: first, 19% of patients with endocarditis seems quite exaggerated to me. If true, it means that in the vast majority this would remain completely asymptomatic and clinically silent. Second, I think that an aggressive screening should be recommended only if proven effective. Otherwise, recommending screening would only increase the number of echo exams, with potentially useless and time- and resource-consuming exams. I think that this paragraph should be at least reformulated or - maybe better - deleted.
Answer 4: We thank the Reviewer for the professional comment. To answer the first point, 19% is actually the published data that can be seen. As far as the second comment, in order to avoid confusion and oversimplified recommendations, we agree with the Reviewer and deleted the paragraph.
Comment 5: Table 3: the rows "Lung vs Other cancers" are completely useless (the information is already described in the columns)
Answer 5: Thanks for your comment. We deleted that.

Round 2
Reviewer 1 Report
The authors have revised the manuscript according to the comment of the reviewer and the manuscript is more acceptable.